# Chinese CO$_2$ emission flows have reversed since the global financial crisis

Zhifu Mi [1,2], Jing Meng[3], Dabo Guan [1], Yuli Shan [], Malin Song[4], Yi-Ming Wei[5,6], Zhu Liu [3] & Klaus Hubacek [7,8]

This study seeks to estimate the carbon implications of recent changes in China's economic development patterns and role in global trade in the post-financial-crisis era. We utilised the latest socioeconomic datasets to compile China's 2012 multiregional input-output (MRIO) table. Environmentally extended input-output analysis and structural decomposition analysis (SDA) were applied to investigate the driving forces behind changes in CO$_2$ emissions embodied in China's domestic and foreign trade from 2007 to 2012. Here we show that emission flow patterns have changed greatly in both domestic and foreign trade since the financial crisis. Some economically less developed regions, such as Southwest China, have shifted from being a net emission exporter to being a net emission importer. In terms of foreign trade, emissions embodied in China's exports declined from 2007 to 2012 mainly due to changes in production structure and efficiency gains, while developing countries became the major destination of China's export emissions.

[1] Water Security Research Centre, School of International Development, University of East Anglia, Norwich, NR4 7TJ, UK. [2] The Bartlett School of Construction and Project Management, University College London, London, WC1E 7HB, UK. [3] School of Environmental Sciences University of East Anglia, Norwich NR4 7TJ, UK. [4] School of Statistics and Applied Mathematics Anhui University of Finance and Economics, Bengbu 233030, China. [5] Center for Energy and Environmental Policy Research Beijing Institute of Technology, Beijing 100081, China. [6] School of Management and Economics Beijing Institute of Technology, Beijing, 100081, China. [7] Department of Geographical Sciences, University of Maryland, College Park, MD 20742, USA. [8] Department of Environmental Studies, Masaryk University, Brno, 602 00, Czech Republic. Zhifu Mi and Jing Meng contributed equally to this work. Correspondence and requests for materials should be addressed to D.G. (email: dabo.guan@uea.ac.uk)

In 2006, China became the largest emitter of carbon dioxide ($CO_2$) emissions in the world[1, 2]. China's economy was affected by the 2008 global financial crisis, resulting in a gradual decline in economic growth. Although its economy has been recovering slowly, China has been unable to continue the rapid economic growth that occurred before the recession. Instead, the country has entered a new phase of economic development—a new normal—in which its economic development mode has changed greatly.

First, China's production and consumption structure has changed considerably[3, 4]. The share of investment peaked in 2009, accounting for 80% of gross domestic product (GDP) growth. Since 2009, the share of consumption has steadily increased, accounting for approximately 65% of GDP growth in 2015. With these structural changes, the associated share of $CO_2$ emissions has also changed, with consumption-related emissions having increased since 2010[5].

Second, we observe a new domestic division of labour reflected in the fact that the economy is growing faster in western China than in eastern China. Although China is usually viewed as a homogenous entity in climate change research, it is a vast country with great regional variations in economic development, resource endowments, population and lifestyles. Specifically, the considerable economic gap between western and eastern China has recently begun to close. Large amounts of $CO_2$ are emitted in poorer western regions to support consumption and exports in the richer eastern regions[6–8]. China has made great efforts to balance economic development among the provinces and to narrow the gap between the east and the west. In 2000, China launched the Western Development Strategy, which calls for the government to invest in infrastructure development and natural resource exploitation and sets preferential policies to create regional economic development centres for China's western region. The strategy has entered its second phase (i.e. 2010–2030), during which western provinces are expected to achieve higher rates of economic growth. For example, the growth rate of fixed asset investment is much larger in western China than in the developed eastern parts of the country. As a result, the per capita consumption and GDP growth has been much faster in western China than in eastern China since the global financial crisis.

Third, China's role in international trade has changed since the global financial crisis. The global economy has been marked by slow growth and sluggish trade since 2008. Although China's foreign trade growth has declined considerably compared to pre-crisis levels[9], it is still faster than growth in global trade. However, the driving forces behind China's foreign trade are now different from those prevailing before the financial crisis[3]. Su and Thomson[10] estimated China's emissions embodied in both processing and normal exports during 2006–2012 and applied structural decomposition analysis (SDA) to identify the driving forces to the embodied emission changes. Their results show that the driving forces changed greatly after the financial crisis. China's participation in the global value chain is moving upstream. The share of domestic value added in China's exports has increased since 2005, suggesting that China's exports are moving to higher value-added products rather than cheap products[11].

Fourth, the geographic patterns in China's foreign trade have also changed. China's exports were previously highly dependent on the import demand from developed economies, especially the United States and European markets. After the 2008 crisis, however, the import demand from developing countries grew stronger than that from developed countries. In other words, the dominance of developed economies in China's foreign trade has declined, while trade with developing countries increased[3].

The patterns of emissions embodied in China's domestic and foreign trade have changed since the economic recession but the interregional carbon emission flows in China and internationally in the post-financial-crisis era have not been analysed thoroughly. This study shows these new carbon flow patterns within China and analyses the domestic drivers as well as investigates China's new role in global trade as reflected by embodied carbon emission flows between countries. We developed a Chinese multi-regional input-output (MRIO) dataset and analysed the recent changes in interregional emission flows and regional carbon emissions. We applied SDA to investigate the driving forces behind these changes in emissions embodied in trade.

Details of the construction of the MRIO dataset, environmentally extended input-output analysis (EEIOA) and SDA model[12, 13] are described in the Methods section. In brief, we constructed a Chinese MRIO table for 30 provinces and 30 sectors. The Chinese MRIO table was linked to a global MRIO model based on the Global Trade and Analysis Project (GTAP) database. We then used EEIOA to estimate and compare the changes in $CO_2$ emissions embodied in China's domestic trade between 30 provinces and those embodied in each province's exports to other countries. The SDA model was further applied to investigate the driving forces behind these changes in flows of embodied emissions. Our findings show that, during 2007–2012, in terms of domestic trade, those economically less developed regions, such as Southwest China, have shifted from being a net emission exporter to being a net emission importer. China's international emission trade destinations have partially shifted from developed countries to developing countries, due to significant increase of consumption in emerging economies.

## Results

**Interregional carbon emission flows in 2012**. In China, large amounts of $CO_2$ emissions related to goods and services consumed in the richer eastern coastal provinces are imported from poorer provinces in Central and Western China. In 2012, approximately 50% of $CO_2$ emissions in China were emitted during the production of goods and services that were ultimately consumed in different provinces in China or abroad. Based on the results of the 30-province MRIO, we aggregate the results for eight regions (i.e. Northeast, Beijing–Tianjin, North, Central, Central Coast, South Coast, Northwest and Southwest) frequently used in analyses for comparability and ease of presentation and discussion.

Among the eight regions, Northwest China had the largest emission outflows (Supplementary Fig. 1). The $CO_2$ emission outflows from this region were 533 million tonnes (Mt) in 2012, accounting for 22% of the total domestic trade-related emissions. As the least developed region in the country, Northwest region supports the economic development of Central and Eastern regions by providing high-carbon-intensive and low-value-added products. Large proportions of the $CO_2$ emissions in Northwest region are induced by the production of goods and services consumed in developed regions in China. More than 55% of $CO_2$ emissions in Northwest region were emitted during the production of goods that were ultimately consumed in other regions or abroad in 2012. For example, Northwest region had 120 and 93 Mt $CO_2$ emissions embodied in exports to Central and Central Coast regions, respectively.

The eastern coastal provinces (including Beijing–Tianjin, Central Coast and South Coast regions) are the most affluent regions in China, with large emission inflows from poorer central and western China. More than 70% of the $CO_2$ emissions embodied in the goods and services consumed in Beijing–Tianjin originated from other regions in 2012, and approximately 50% of the $CO_2$ emissions embodied in the goods and services consumed in Central Coast and South Coast regions originated in other

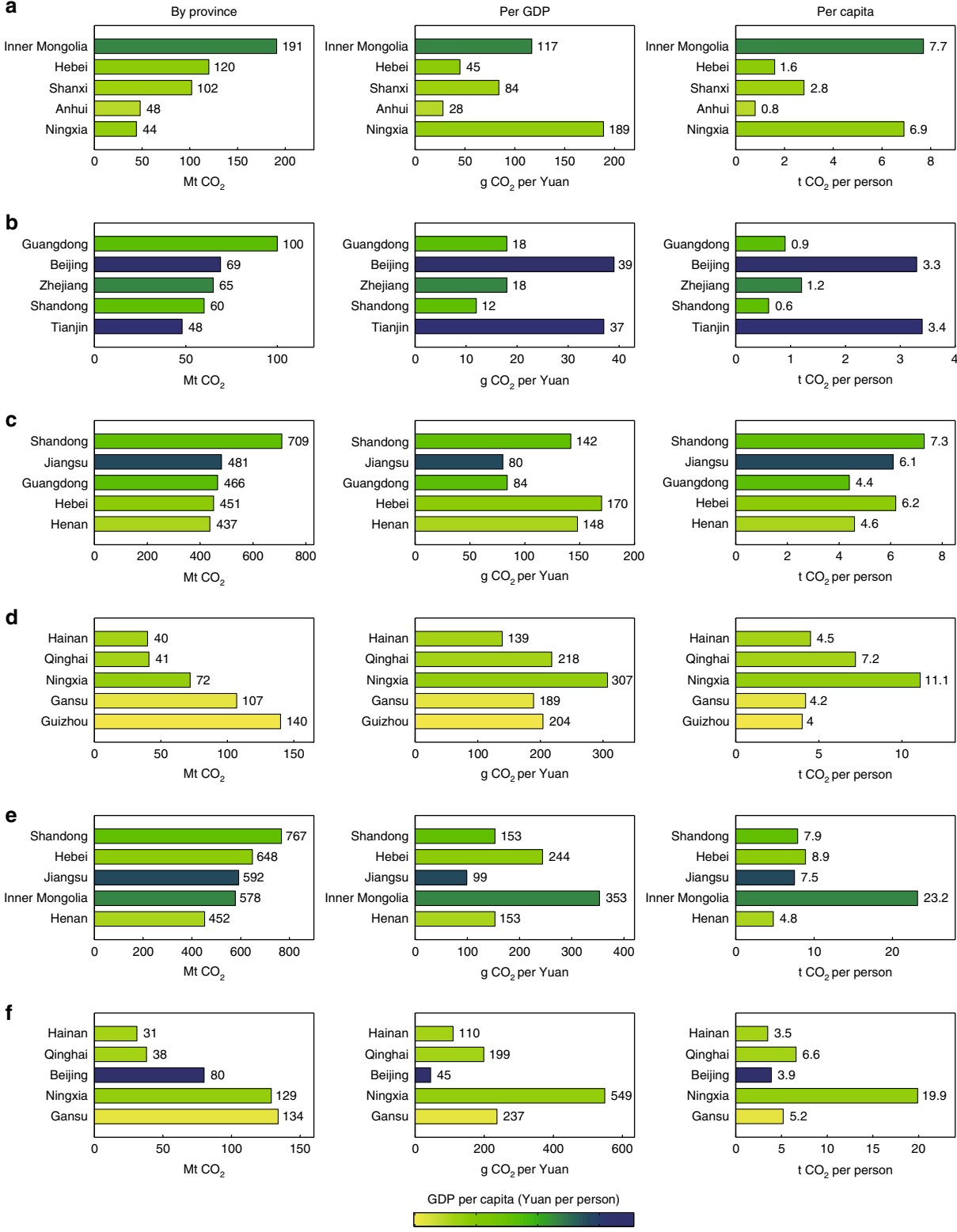

**Fig. 1** Comparisons of emissions and flows at provincial level in China in 2012. **a** Top 5 net domestic emission outflows, **b** Top 5 net domestic emission inflows, **c** Top 5 consumption emissions, **d** Bottom 5 consumption emissions, **e** Top 5 territory emissions and **f** Bottom 5 territory emissions. The colours of bars correspond to provincial gross domestic product (GDP) per capita, from the most affluent provinces in red to the least developed provinces in green (see the scale). Mt means million tonnes, g means gram, t means tonne and $CO_2$ means carbon dioxide. See Supplementary Table 1 for territory-based and consumption-based $CO_2$ emissions for China's 30 provinces

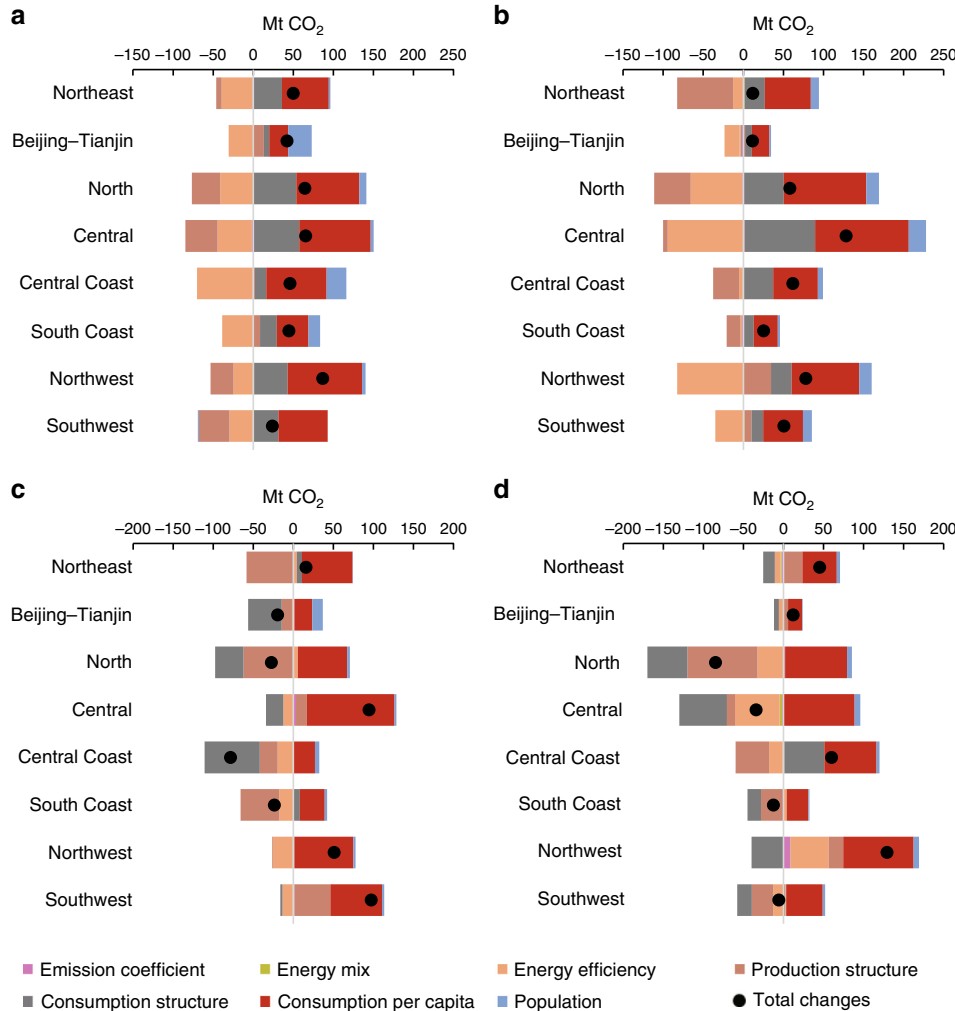

**Fig. 2** Contributions of different factors to changes in domestic inflow and outflow in China. **a** Inflow changes from 2007 to 2010, **b** Outflow changes from 2007 to 2010, **c** Inflow changes from 2010 to 2012 and **d** Outflow changes from 2010 to 2012. Mt $CO_2$ means million tonnes of carbon dioxide. Emission coefficient is carbon emissions per unit of energy consumption, and energy efficiency is energy consumption per unit of total output

regions. As the capital of China, Beijing outsourced more than 85% of its emissions to other regions. For example, Beijing imported 20 and 21 Mt $CO_2$ emissions from Hebei and Inner Mongolia in 2012, accounting for 14 and 15% of its outsourced emissions, respectively.

The provinces with the largest domestic net emission outflows are mainly in poorer western China (Fig. 1a, by province), while those with the largest domestic net emission inflows are mainly located in richer eastern regions (Fig. 1b, by province). The western provinces mainly export high-carbon-intensive products while importing low-carbon-intensive products. Inner Mongolia is the province with the largest net emission outflows because it is one of the key providers of energy products in China. For example, Inner Mongolia provides large amounts of electricity to neighbouring regions, with 133 billion kilowatt hours (kW·h) in net electricity exports in 2012[14, 15]. As a result, its total $CO_2$ emissions per unit of domestic exports were 186 g per Yuan in 2012, which were approximately three times that of its domestic imports (59 g per Yuan). In comparison, the eastern provinces usually import high-carbon-intensive products while producing and exporting low-carbon-intensive products. Guangdong had the largest net domestic emission inflows. The $CO_2$ emissions per unit of Guangdong's domestic imports were 64 g per Yuan, which were approximately two times that of its domestic exports (35 g per Yuan).

Total consumption-based $CO_2$ emissions are greatest in affluent eastern provinces, such as Shandong, Jiangsu and Guangdong, as well as in populous provinces, such as Hebei and Henan (Fig. 1c, by province). In comparison, the provinces with the lowest consumption-based emissions are mainly the least developed provinces in western regions, such as Qinghai, Ningxia and Gansu, as well as provinces with small populations, such as Hainan (Fig. 1d, by province). The consumption-based emission intensity (i.e. the consumption-based $CO_2$ emissions per unit of GDP) in eastern provinces is usually smaller (Fig. 1c, per GDP, and Fig. 1d, per GDP) because of the lower proportion of energy- and carbon-intensive products throughout the supply chain. For example, the consumption-based emission intensity was 80 and 84 g per Yuan in Jiangsu and Guangdong, respectively, which are two of the richest provinces in eastern China. In comparison, Ningxia had the largest consumption-based emission intensity in 2012, which was about four times of that in Jiangsu.

Total territory-based $CO_2$ emissions are the largest in large developed provinces, such as Shandong and Jiangsu, as well as in provinces whose economies depend on heavy industry, such as Hebei and Inner Mongolia (Fig. 1e, by province). By contrast, provinces with the smallest total territory-based $CO_2$ emissions are mainly the least developed western provinces, such as Qinghai, Ningxia and Gansu, as well as those provinces with a higher proportion of services in their industry structure, such as

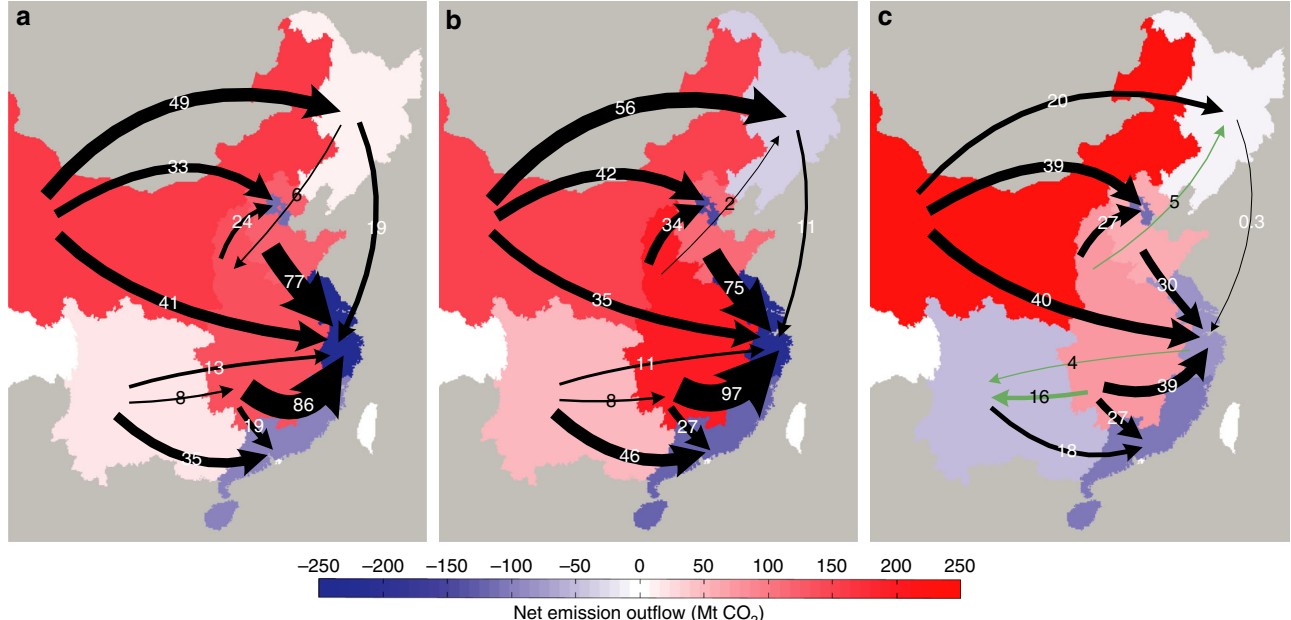

**Fig. 3** Changes in emission flow patterns within China. **a** Net emission outflows in 2007, **b** Net emission outflows in 2010, **c** Net emission outflows in 2012. The sizes of arrows correspond to the amount of net emission flows, and green arrows mean that net emission outflows reversed between 2007 and 2012. Mt $CO_2$ means million tonnes of carbon dioxide. The net emission flows to eastern coastal provinces declined from 2007 to 2012, and Southwest and Northeast China shifted from being net emission exporters to net emission importers

Beijing and Hainan (Fig. 1f, by province). Territory-based emissions are usually higher than consumption-based emissions in less developed provinces, while consumption-based emissions are higher in developed regions. For example, the total territory-based emissions of Inner Mongolia, one of key energy providers in western China, was 578 Mt in 2012, a figure that is approximately two times that of its consumption-based emissions. As a result, Inner Mongolia had relatively high per GDP and per capita territory-based emissions (Fig. 1e, per GDP and capita). By contrast, the total consumption-based $CO_2$ emissions of Beijing, the capital of China, was approximately two times the total territory-based emissions in 2012. As a result, the per GDP and per capita product-based emissions were 45 g per Yuan and 3.9 tonne per person, respectively, in 2012 (Fig. 1f, per GDP and capita), which were smaller than the values calculated using consumption-based accounting. Therefore, great differences between consumption- and territory-based accounting arise in terms of total, per GDP and per capita emissions.

**Three features in the emission flow patterns within China.** Generally, the eastern coastal regions outsource $CO_2$ emissions to central and western regions in China. Chinese carbon emission flows changed from 2007 to 2012 because of the changes in Chinese production and consumption patterns. First, the provincial-level emission outsourcing rate declined from 2007 to 2012. The outsourcing rate is defined as the share of $CO_2$ emissions that are emitted during the production of goods and services outside a region (in other words, the imported embodied emissions) in comparison to the total consumption-based emissions of a region. For example, 71% of the $CO_2$ emissions embodied in goods and services consumed in Jilin, a province in Northeast China, were imported from other provinces or from abroad in 2007, but this figure declined to 47% in 2012.

Second, the net emission flows to eastern coastal provinces declined from 2007 to 2012, especially for Central Coast region. The net emission flows to Central Coast region declined by 66%

from 2007 to 2012. To be specific, the net emission flows to the three provinces in Central Coast region, i.e., Shanghai, Zhejiang and Jiangsu, declined by 91%, 41% and 45% from 2007 to 2012, respectively. The emission inflows of Central Coast region declined by 8% from 2007 to 2012 mainly due to changes in production and consumption structure. The two factors would have driven an increase in emission inflows by 5% and 13%, respectively, with other factors held constant (Fig. 2). In comparison, the emission outflows of Central Coast region increased by 63% from 2007 to 2012 mainly due to changes in consumption structure and the level of consumption per capita. The two factors would have offset emission outflows by 45% and 63%, respectively (Fig. 2). From a sectoral perspective, the emissions embodied in the net inflows in the electricity and hot water production of Central Coast region declined from 78 to 3 Mt in 2007–2012 (Supplementary Fig. 2). The emissions embodied in the net inflows in the metallurgy sector of Central Coast region also declined by 55% during the same period. From a regional perspective, the $CO_2$ emissions embodied in the net trade from North and Central regions to Central Coast region declined by 61% and 55%, respectively (Fig. 3). In comparison, the emissions embodied in the net trade from Central Coast region to Southwest region changed from −13 to 4 Mt.

Third, Southwest and Northeast China shifted from being net emission exporters in 2007 to net emission importers in 2012, mainly due to the rapid growth of consumption in these regions. Southwest region was a net emission exporter in 2007, with 22 Mt net emission outflows. However, this region became a net emission importer in 2012 with 54 Mt net emission inflows. The growth in consumption per capita in Southwest region would have driven its emission inflows increase by 76% from 2007 to 2012 with other factors held constant (Fig. 2). From a sectoral perspective, the emissions embodied in the net outflows of electricity and hot water production in Southwest region were 20 Mt in 2007, changing to net inflows of 59 Mt in 2012 (Supplementary Fig. 2). From a regional perspective, the net emission outflows of Southwest region to Central and Central

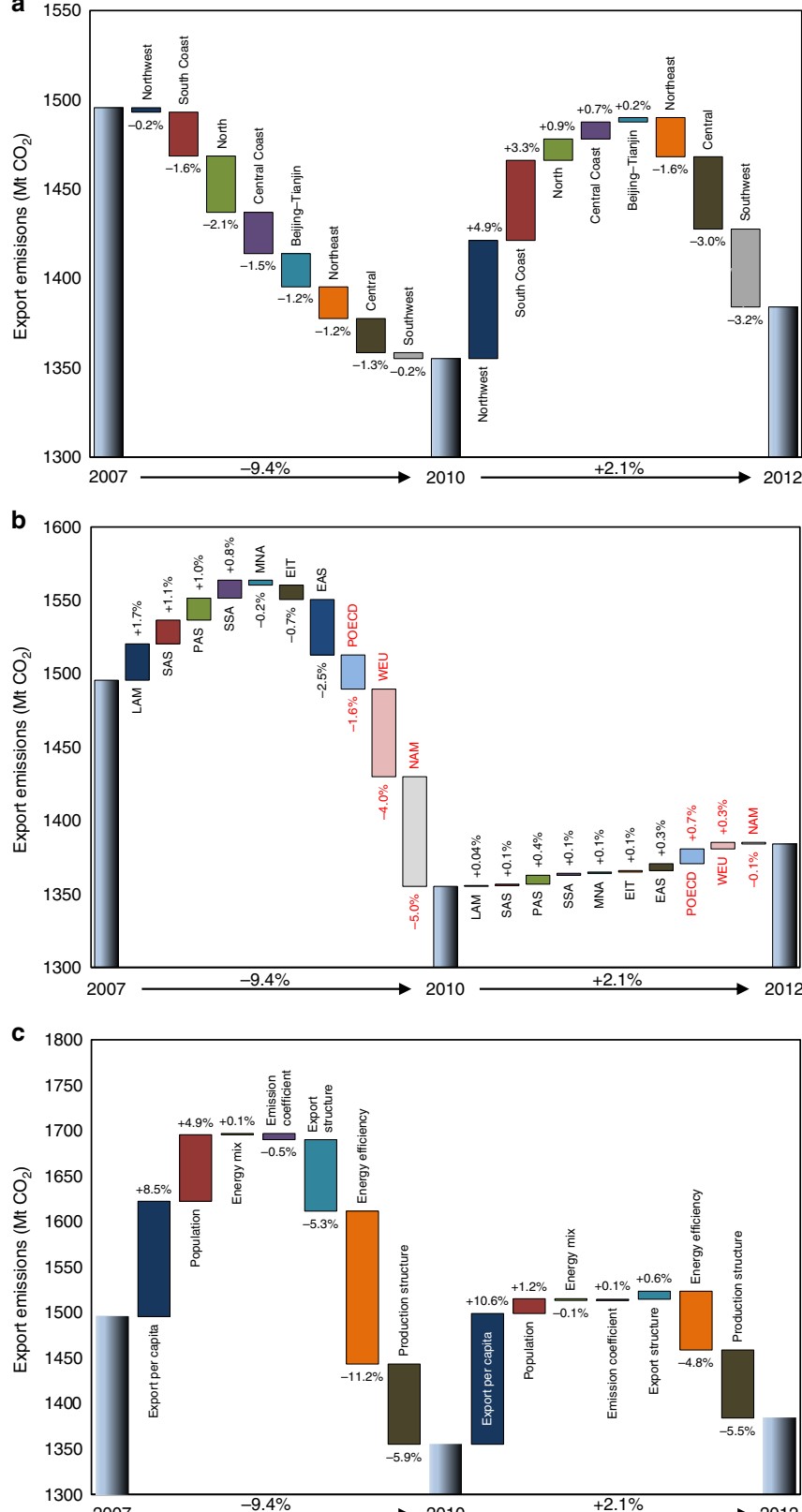

**Fig. 4** Changes in emissions embodied in exports between 2007 and 2012. **a** Changes in the export emissions from eight Chinese regions. **b** Changes in China's export emissions to 10 global regions. Regions in red are developed countries, while regions in black are developing countries. LAM: Latin America and the Caribbean; SAS: South Asia (mainly India); PAS: South-East Asia and the Pacific; SSA: Sub-Saharan Africa; MNA: the Middle East and North Africa; EIT: Economies in Transition (including Eastern Europe and the Former Soviet Union); POECD: Pacific OECD-1990 Countries (including Japan, Australia, and New Zealand); EAS: East Asia (China excluded); WEU: Western Europe; NAM: North America (USA and Canada). See Supplementary Data 1 for definitions of world regions. **c** Contributions of different factors to changes in China's export emissions. Mt $CO_2$ means million tonnes of carbon dioxide

Coast regions were 8 and 13 Mt in 2007 and changed to net inflows of 16 and 4 Mt in 2012, respectively (Fig. 3). As a net emission exporter with 3 Mt net emission outflows in 2007, the Northeast became a net emission importer with 6 Mt net emission inflows in 2012. Growth in consumption per capita in the Northeast would have increased emission inflows by 65% from 2007 to 2012. In comparison, changes in production structure would have offset emission outflows by 24% from 2007 to 2012 (Fig. 2). From a regional perspective, emissions embodied in net trade outflows from the Northeast to the Central and Central Coast regions were 6 and 19 Mt in 2007, respectively, which changed to −5 and 0.3 Mt in 2012, respectively (Fig. 3).

**Chinese export emissions in international trade**. The emissions embodied in China's exports declined from 2007 to 2012. China's export emission levels declined by 9.4% from 2007 to 2010 and have not returned to the pre-crisis levels, although emissions related to exports increased by 2.1% from 2010 to 2012 (Fig. 4). The trend of changes in emissions embodied in China's exports is consistent with the results of Su and Thomson[10]. Exports were an important driver of China's economic growth but also contributed significantly to its increase in $CO_2$ emissions. However, China's export growth declined sharply after the 2008 global financial crisis, and as a result, the $CO_2$ emissions embodied in exports from 2007 to 2010 also declined[16, 17]. For instance, the $CO_2$ emissions embodied in the exports of metal and textile products both declined by 43% and 29% from 2007 to 2012, respectively.

Export emissions in most Chinese regions, except in Northwest and South Coast regions, declined from 2007 to 2012. For the period 2007–2010, export emissions in all eight regions declined. The regions with the largest reductions in export emissions were mainly in eastern China, where exports played a critical role in economic development (Fig. 4). For instance, the export emissions of Shanghai and Beijing, the two largest cities in China, declined by 34% and 30%, respectively, from 2007 to 2010. China's export emissions grew slightly from 2010 to 2012, with the western regions contributing the largest share to emission growth. The export emissions in three regions, i.e., Northeast, Central and Southwest, continued to decline, offsetting China's export emissions by 1.6%, 3.0% and 3.2%, respectively, from 2010 to 2012 (Fig. 4). In comparison, the emissions embodied in Northwest China's exports increased by 66 Mt $CO_2$ during the same period, leading to a 4.9% growth in China's export emissions. For example, the export emissions of the north western regions Xinjiang and Shaanxi increased by 29% and 33%, respectively, from 2010 to 2012.

The destinations of China's export emissions have partially shifted from developed countries to developing countries. China's emission exports to developed countries declined from 2007 to 2010, while emission exports to most developing countries grew. For the period 2010–2012, more than half of China's export emissions resulted from the growth in foreign trade to developing countries. Before this change, China's exports were highly dependent on the import demand from developed economies, especially the United States and European markets. The emissions exported from China to developed countries accounted for more than 60% of the total overall export emissions in 2007. For example, the emissions levels embodied in exports to North America and Western Europe were 377 and 351 Mt $CO_2$ in 2007, accounting for 25 and 23% of the total, respectively. After the global financial crisis, however, the import demand from developed economies was much weaker than that from developing economies. Emissions from China's exports to North America and Western Europe declined by 20% and 16%, respectively, from

2007 to 2012 (Fig. 4). By contrast, emissions from China's exports to developing regions increased by 6% from 2007 to 2012, mainly as a result of the growth in South-South trade[18, 19]. Emissions from China's exports to Latin America and the Caribbean increased by 33% from 2007 to 2012. For example, emissions from China's exports to Brazil increased by 63% from 2007 to 2012. The emissions embodied in exports to South Asia and to South-East Asia and the Pacific also grew by 30% and 25%, respectively, during the same period. For instance, emissions from China's exports to India grew by 36% from 2007 to 2012. As a result, the proportion of emissions embodied in exports to developing countries in China's overall export emissions increased from 40% in 2007 to 46% in 2012.

The decline of China's export emissions from 2007 to 2012 was mainly due to production structure changes and energy efficiency gains. Growth in export volume (i.e. the product of export per capita and population) was the strongest factor which drove China's increase in export emissions. The growth of export per capita would have increased export emissions by 8.5% from 2007 to 2010 and 10.6% from 2010 to 2012 with other factors held constant (Fig. 4). From 2007 to 2010, energy efficiency gains was the strongest factor to offset China's export emissions. It would have decreased export emissions by 11.2% from 2007 to 2010 (Fig. 4). From 2010 to 2012, production structure changes surpassed energy efficiency gains becoming the strongest factor to offset China's export emissions. Production structure changes and energy efficiency gains would have decreased export emissions by 5.5 and 4.8% from 2010 to 2012, respectively. Above results show that production structure changes were an important factor to offset China's export emissions during 2007–2012. In the new normal, China has been making efforts to change its economic development mode and targeting high-quality economic growth, i.e., growth driven by higher value added and lower resource intensive inputs. The country is taking numerous measures to save energy, reduce carbon emissions and control local air pollution. Therefore, production structure changes are expected to continue to decrease (export) emissions in the future.

## Discussion

Great imbalances in $CO_2$ emissions embodied in domestic trade are a reflection of the discrepancies in the levels of economic development between provinces. Generally, large amounts of $CO_2$ emissions related to goods and services consumed in highly developed eastern coastal provinces are imported from less developed provinces in central and western China. The affluent eastern coastal provinces import predominantly low-value-added and high-carbon-intensive products from less developed provinces in China, while exporting high-value-added and low-carbon-intensive products. In other words, consumption in the richer eastern regions is supported by emissions occurring in the poorer central and western parts of China[6].

The carbon emissions embodied in domestic trade have changed considerably in China because of changing patterns of regional economic growth. Net emission flows to eastern coastal provinces declined from 2007 to 2012, while some less developed regions (such as the Southwest and Northeast) shifted from being net emission exporters in 2007 to net emission importers in 2012. This shift was mainly due to the rapid growth of consumption volume in western China and changes in production structure. The net emission flows from western regions to eastern regions in China may further decline because of the faster economic growth in the western regions. China is struggling to balance economic development among provinces and to narrow the gap between the East and the West. The country has adopted many preferential policies to boost the economies of the western provinces.

For example, China's Western Development Strategy has entered its second phase (i.e. 2010–2030), during which more investments will be devoted to infrastructure development and natural resource exploitation in western regions. Additionally, the Chinese government has proposed the Silk Road Economic Belt and the twenty-first-century Maritime Silk Road development strategy, also known as The Belt and Road (B&R). Under the B&R strategy, trade between the western Chinese regions and the rest of Eurasia will be promoted. Under this plan, foreign exports from the western regions are supposed to increase, especially to countries in the Silk Road Economic Belt. Therefore, production for domestic and foreign consumption would grow much faster in the western regions than in the eastern regions. Accordingly, there would be a further reversal of the direction of emissions embodied in the trade between western and eastern China.

China's role in global supply chains has also been experiencing a major structural shift. The emissions embodied in China's foreign exports declined from 2007 to 2012, mainly due to production structure changes and efficiency gains. The destinations of China's export emissions have been shifting from developed countries to developing countries. Emissions embodied in China's exports to North America and Western Europe declined by 20% and 16%, respectively, while those embodied in exports to Latin America and the Caribbean increased by 33% from 2007 to 2012.

The recent trajectory implies that the destinations of China's foreign export emissions would further shift from developed countries to developing countries because of China's changing role in global trade. The global economy has been marked by slow growth and sluggish trade since the global financial crisis. However, the import demand from developing countries has grown stronger than that from developed countries. The share of developing economies in international trade has been increasing mainly because of the rapid development in South-South trade, i.e., trade with and among developing countries. China's increasing volume of trade with other developing countries contributes greatly to the development of South-South trade. China has been increasing its investments in emerging economies and promoting trade with developing countries. For example, China's foreign direct investment in Africa reached US$ 32 billion in 2014, with more than 3000 Chinese firms in Africa[20]. More opportunities to advance trade between China and Africa will arise under the B&R strategy and Agenda 2063, which is a 50-year development plan to build an integrated, prosperous and peaceful Africa[21]. Therefore, emissions from China's exports to developing economies will further increase.

In recent years, many researchers have proposed that consumption-based accounting be applied to re-allocate the responsibilities of mitigating climate change because of the large net emission flows from developing countries to developed countries[22, 23]. China has made a dominant contribution to these net emission flows, but emissions embodied in its exports to developed countries have declined. Consumption-based accounting-related policies that address carbon leakage between developed and developing countries are less relevant. Outsourcing of carbon emissions is a global problem not only between developed and developing countries, but increasingly between developing countries. We need to pay more attention to the $CO_2$ emission transfers among developing countries because of the rapid development of South-South trade.

## Methods
**Construction of the China MRIO tables**. We compiled a MRIO database for China's 26 provinces and 4 cities, except Tibet and Taiwan (in total, 30 regions). These regions are divided into eight Chinese regions: Northeast (Heilongjiang, Jilin, Liaoning), Beijing–Tianjin (Beijing, Tianjin), North (Hebei, Shandong), Central

(Henan, Shanxi, Anhui, Hunan, Hubei, Jiangxi), Central Coast (Shanghai, Zhejiang, Jiangsu), South Coast (Guangdong, Fujian, Hainan), Northwest (Inner Mongolia, Shaanxi, Gansu, Ningxia, Qinghai, Xinjiang) and Southwest (Sichuan, Chongqing, Yunnan, Guizhou, Guangxi).

The Chinese MRIO model is compiled based on the input-output tables (IOTs) for 30 Chinese provinces that are published by the National Statistics Bureau. There are 42 economic sectors and five final demand categories, including rural household consumption, urban household consumption, government consumption, fixed capital formation and changes in inventories. Exports are divided into international and domestic exports, and imports are divided into international and domestic imports. We aggregate the provincial IOTs into 30 sectors because of data unavailability (see Supplementary Data 2 for the concordance of sectors). We use the gravity model and modify it with interactions among different regions for the same sector[24]. In the standard gravity model, the interregional trade flows are specified as a function of the total regional outflows, total regional inflows and transferring cost, which is usually proxied by a distance function. The gravity model is

$$y_i^{rs} = e^{\beta_0} \frac{(x_i^{rO})^{\beta_1}(x_i^{Os})^{\beta_2}}{(d^{rs})^{\beta_3}}, \tag{1}$$

where $y_i^{rs}$ is the trade flows of sector $i$ from region $r$ to region $s$, $e^{\beta_0}$ is the constant proportionality, $x_i^{rO}$ is the total outflows of sector $i$ from region $r$, $x_i^{Os}$ is the total inflows of sector $i$ to region $s$, $d^{rs}$ is the distance between region $r$ and region $s$ (we use the distance between the capital cities of the two provinces in the study), $\beta_1$ and $\beta_2$ are weights assigned to the masses of origin and destination, respectively, and $\beta_3$ is the distance decay parameter. Equation (1) can be transformed into

$$\ln(y_i^{rs}) = \beta_0 + \beta_1 \ln(x_i^{rO}) + \beta_2 \ln(x_i^{Os}) - \beta_3 \ln(d^{rs}) + \varepsilon \tag{2}$$

and further into

$$\mathbf{Y} = \beta_0 \mathbf{L}_0 + \beta_1 \mathbf{X}_1 + \beta_2 \mathbf{X}_2 - \beta_3 \mathbf{X}_3 + \boldsymbol{\varepsilon} \tag{3}$$

where $\mathbf{Y}$ is a $N \times 1$ matrix that represents the logarithm of the trade flows of product $i$ between regions, $\mathbf{L}_0$ is a $N \times 1$ matrix with all elements equal to 1, $\mathbf{X}_1$ and $\mathbf{X}_2$ are the logarithm of the total outflows from origin regions and total inflows to destination regions, respectively, and $\mathbf{X}_3$ is the logarithm of the distance between two regions. Equation (3) can be solved by using multiple regression.

There are different interregional competition and cooperation relationships for different sectors. The industrial supply chains in some sectors are shorter, and there may be competitive relationships among different regions for these sectors, such as agriculture, food processing and textiles. In comparison, the industrial supply chains in other sectors are longer, and there may be more cooperative relationships among different regions for these sectors, such as machinery and chemicals. To reflect interregional competition and cooperation in our analysis, we introduce the concept of impact coefficients among different regions for the same sector. The impact coefficient for one sector is obtained by

$$\begin{cases} c_i^{gh} = \frac{\mu_i^g + \mu_i^h}{|\mu_i^g - \mu_i^h| + \min\limits_{r=1,2,\ldots,n} \mu_i^r} & g \neq h \\ c_i^{gh} = 1 & g = h \end{cases} \tag{4}$$

where $c_i^{gh}$ is the impact coefficient between regions $g$ and $h$ for sector $i$, $\mu_i^g$ and $\mu_i^h$ are the location entropy of sector $i$ in regions $g$ and $h$, respectively, and $n$ is the number of regions. The impact coefficients indicate that stronger interactions for sector $i$ occur between regions $g$ and $h$ if the location entropy of the sector in both regions is higher. Equation (4) indicates that $c_i^{gh} > 1$ when $g \neq h$ and that a larger figure means stronger interactions. In addition, $c_i^{gh} = 1$ when $g = h$.

We also introduce the concept of impact exponents among different regions for the same sector. It is assumed that if a larger proportion of one sector's output is used for its own intermediate inputs, then interregional cooperation exists for the sector. The impact exponent for one sector is obtained by

$$\theta_i = \overline{\delta} - \delta_i \tag{5}$$

where $\theta_i$ is the impact exponent for sector $i$, $\delta_i$ is the proportion of the total output of sector $i$ that it uses as its own intermediate inputs and $\overline{\delta}$ is the average value of $\delta_i$. If $\theta_i > 0$, there are competitive relationships for sector $i$. Otherwise, there are cooperative relationships for sector $i$.

We use the impact coefficients and impact exponents to modify the interregional trade flows that are obtained by the standard gravity model. The formula is

$$Y' = \hat{Y} / \left(c_i^{gh}\right)^{\theta_i} \tag{6}$$

where $Y'$ represents the modified trade flows of sector $i$ and $\widehat{Y}$ represents the trade flows, which are obtained by the standard gravity model.

The initial trade flow matrix produced above, which excludes intraregional flows, does not meet the double sum constraints, in which the row and column totals match the known values in the 2012 IOTs. We use the RAS approach to

adjust the trade flow matrix to ensure agreement with the sum constraints[25]. The RAS approach tends to preserve the structure of the initial matrix as much as possible with a minimum number of necessary changes to restore the row and column sums to the known values[26].

In addition to the provincial IOTs, China also published a national IOT for 2012. There are great gaps between the national IOT and provincial IOTs. For example, the sum of the total output of the 30 provinces in the provincial IOTs is 7% higher than the national total output in the national IOT. The total amount in the national IOT is assumed to be more accurate, while provincial IOTs more closely represent the economic structure at the provincial level. Therefore, we use the national IOT to adjust the total amount of output, value added, and international export and import in the MRIO table which is compiled based on provincial IOTs.

**Linking the Chinese MRIO to the GTAP database.** The Chinese MRIO models are connected to global MRIO models which are based on version 9 of the GTAP database[27]. Similar linkages have been done by previous studies. For example, Feng et al.[6] and Weitzel and Ma[8] both linked the 2007 China MRIO table with the 2007 GTAP database, and Su and Ang[7] linked the 1997 China MRIO table with the 2000 Asian international IOT. The GTAP database describes international trade connections for 57 economic sectors among 129 regions in 2007 and 140 regions in 2011. Because the Chinese MRIO models are for 2007, 2010 and 2012, we update the economic and emission performance of the world economics in the 2011 GTAP model to the year 2010 and 2012. All IO tables are deflated to 2012 prices using the double deflation method[28]. See Supplementary Table 2 for more details on the double deflation method.

Because China is one of the GTAP regions, we disaggregate the Chinese IOT in the GTAP data into 30-region and 30-sector tables according to our Chinese MRIO models. The 57-sector international import and export matrices for China in the GTAP data are aggregated to 30 sectors (see Supplementary Data 3 for the concordance of sectors). Based on the adjusted GTAP international import and export matrices for China, the international Chinese imports and exports for each sector and each province are disaggregated by country (128 regions for 2007 and 139 regions for 2010 and 2012). The international exports (or imports) of a sector in a province are assumed to be distributed among all foreign countries in the same proportion as China's exports (or imports) for that sector. The new global MRIO then includes 30 Chinese provinces and 128 (or 139) countries with 30 sectors for Chinese provinces and 57 sectors for foreign countries. For the final demand, there are five sectors for Chinese provinces (rural household consumption, urban household consumption, government consumption, fixed capital formation and changes in inventories) and three sectors for foreign countries (investment, household consumption and government consumption). See Supplementary Fig. 3 for more details on linking the Chinese MRIO to the GTAP database.

**$CO_2$ emission inventory construction.** We use the approach provided in the Intergovernmental Panel on Climate Change (IPCC) reference to calculate the $CO_2$ emissions from energy combustion based on China's provincial energy statistics[29], [30]. The calculation formula is

$$C = E \times V \times F \times O, \tag{7}$$

where $C$ is the fossil-fuel-related $CO_2$ emissions, $E$ is the amount of energy consumption from different fuel types (in physical unit), $V$ is the net calorific value of different fuel types, $F$ is the carbon content that represents $CO_2$ emissions when unit heat is released and $O$ is the oxygenation efficiency of different fuel types. To avoid missing or double accounting, we calculate the fossil-fuel consumption as follows:

$$E = \text{Total final consumption} + \text{Input for thermal power} + \text{Input for heating} \\ - \text{Used as chemical material} - \text{Loss} \tag{8}$$

**Environmentally extended input-output analysis.** Different regions are connected through interregional trade in an MRIO table. The basic linear equation of the MRIO model is

$$\mathbf{X} = (\mathbf{I} - \mathbf{A})^{-1}\mathbf{F}, \tag{9}$$

$$\mathbf{X} = \begin{bmatrix} \mathbf{X}^1 \\ \mathbf{X}^2 \\ \vdots \\ \mathbf{X}^n \end{bmatrix}, \mathbf{A} = \begin{bmatrix} \mathbf{A}^{11} & \mathbf{A}^{12} & \cdots & \mathbf{A}^{1n} \\ \mathbf{A}^{21} & \mathbf{A}^{22} & \cdots & \mathbf{A}^{2n} \\ \vdots & \vdots & \ddots & \vdots \\ \mathbf{A}^{n1} & \mathbf{A}^{n2} & \cdots & \mathbf{A}^{nn} \end{bmatrix}, \mathbf{F} = \begin{bmatrix} \mathbf{f}^{11} & \mathbf{f}^{12} & \cdots & \mathbf{f}^{1n} \\ \mathbf{f}^{21} & \mathbf{f}^{22} & \cdots & \mathbf{f}^{2n} \\ \vdots & \vdots & \ddots & \vdots \\ \mathbf{f}^{n1} & \mathbf{f}^{n2} & \cdots & \mathbf{f}^{nn} \end{bmatrix}, \tag{10}$$

where $\mathbf{X} = (X_i^s)$ is the vector of total output and $X_i^s$ is the total output of sector $i$ in region $s$. $\mathbf{I}$ is the identity matrix, and $(\mathbf{I} - \mathbf{A})^{-1}$ is the Leontief inverse matrix. The technical coefficient submatrix $\mathbf{A}^{rs} = (a_{ij}^{rs})$ is given by $a_{ij}^{rs} = z_{ij}^{rs}/x_j^s$, in which $z_{ij}^{rs}$ represents the intersectoral monetary flows from sector $i$ in region $r$ to sector $j$ in

region $s$, and $x_j^s$ is the total output of sector $j$ in region $s$. $\mathbf{F} = (f_i^{rs})$ is the final demand matrix, and $f_i^{rs}$ is the final demand of region $s$ for the goods of sector $i$ from region $r$.

To calculate the $CO_2$ emissions embodied in goods and services, we need to calculate the carbon intensity (i.e., $CO_2$ emissions per unit of economic output)[31], [32]. Carbon emissions can be mathematically expressed as

$$C = \mathbf{K}(\mathbf{I} - \mathbf{A})^{-1}\mathbf{F}, \tag{11}$$

where $C$ is the total emissions embodied in goods and services used for final demand and $\mathbf{K}$ is a vector of the carbon intensity for all economic sectors in all regions.

**Structural decomposition analysis.** SDA is a widely used approach to estimate the drivers of changes in carbon emissions and energy consumption. Su and Ang[33] summarised the SDA studies on energy and emissions which were published before 2010, and Wang et al.[34] summarised related SDA studies which were published during 2010–2015. To estimate the drivers of emission changes, the carbon intensity, i.e., $\mathbf{K}$ in Eq. (11), is further decomposed into emission coefficient ($\mathbf{O}$, i.e., emissions per unit of energy consumption), energy mix ($\mathbf{M}$) and energy efficiency ($\mathbf{T}$, i.e., energy consumption per unit of total output). The final demand, i.e., $\mathbf{F}$ in Eq. (11), is further decomposed into consumption structure ($\mathbf{S}$), consumption per capita ($\mathbf{Q}$) and population ($\mathbf{P}$). $\mathbf{L}$ is the Leontief inverse matrix, defined as $\mathbf{L} = (\mathbf{I} - \mathbf{A})^{-1}$ in Eq. (11). Thus the changes in emissions embodied in trade can be decomposed as

$$\Delta C = (\Delta\mathbf{O})\mathbf{MTLSQP} + \mathbf{O}(\Delta\mathbf{M})\mathbf{TLSQP} + \mathbf{OM}(\Delta\mathbf{T})\mathbf{LSQP} + \mathbf{OMT}(\Delta\mathbf{L})\mathbf{SQP} \\ + \mathbf{OMTL}(\Delta\mathbf{S})\mathbf{QP} + \mathbf{OMTLS}(\Delta\mathbf{Q})\mathbf{P} + \mathbf{OMTLSQ}(\Delta\mathbf{P}) \tag{12}$$

where $\Delta$ represents the change in a factor. Each of seven terms in Eq. (12) denotes the contributions to emission changes which are triggered by one driving force if other variables kept constant. The seven factors in our SDA model have $7! = 5040$ first-order decompositions, and different procedures can lead to different results. Su and Ang[33] summarised four SDA methods and pointed out their pros and cons. Mi et al.[5] took the average of all possible first-order decompositions to address this issue. This approach is too time consuming for our MRIO model. In this study, we follow previous studies[35], [36] and use the average of two polar decompositions. Decomposition is started by changing the first variable first, followed by changing the second and third variables, etc. to get the first polar form. The second polar form is derived in the opposite manner. We take the arithmetic average of the SDA results based on the two polar forms.

**Data sources.** The 2012 Chinese national IOT and the IOTs for each of the 30 provinces are published by the National Statistics Bureau. Based on these Chinese single-region IOTs, we compiled the 2012 Chinese MRIO using the modified gravity model[6]. The 2007 and 2010 Chinese MRIO tables are compiled by the Institute of Geographic Sciences and Natural Resources Research, Chinese Academy of Sciences[37], [38]. The global MRIO tables are obtained from version 9 of the GTAP database, which describes the bilateral trade patterns, production, consumption and intermediate use of commodities and services for 57 sectors among 129 regions in 2007 and 140 regions in 2011[27]. The economic and emission performance of the world economics in the 2011 GTAP model is updated to the year 2010 and 2012. The GDP data is obtained from the National Accounts Main Aggregates Database[39], and the fossil-fuel $CO_2$ emissions are obtained from the Carbon Dioxide Information Analysis Center (CDIAC)[40]. The pricing data for China's IOTs were acquired from the China Statistics Yearbook[41–43], while the pricing data for China's imports and global MRIO tables were obtained from the National Accounts Main Aggregates Database[39]. The energy consumption data used to calculate the $CO_2$ emission inventory were obtained from the China Energy Statistical Yearbooks[44], [45]. We used emission coefficients from our previous research[1, 32, 46]. The coefficients are measured based on 602 coal samples from the 100 largest coal-mining areas in China and are assumed to be more accurate than the IPCC default value.

**Data availability.** The emission factors for different energy types used in calculating $CO_2$ emissions are shown in Supplementary Table 3. All MRIO tables and carbon emission inventories developed and used in this study are provided as Supplementary Data. Supplementary Data 4–6 are $CO_2$ emission inventories for China's 30 provinces for 2007, 2010 and 2012, respectively. Supplementary Data 7 is China's MRIO table. Those data can also be freely downloaded from the China Emission Accounts and Datasets (CEADS) website (http://www.ceads.net/)[32, 47].

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

## Acknowledgements

This study was supported by the National Key R&D Program of China (2016YFA0602604, 2016YFA0602603), the Natural Science Foundation of China (41629501, 71521002, 71642004, 71533005), the UK Economic and Social Research Council (ES/L016028/1) Natural Environment Research Council (NE/N00714X/1), British Academy Grant (AF150310) and Czech Science Foundation under the project VEENEX (GA ČR no. 16-17978S).

## Author contributions

Z.M. and D.G. designed the study. Z.M. performed the analysis and prepared the manuscript. J.M. performed the SDA analysis. Z.M. and J.M. compiled 2012 China MRIO table. Y.S. compiled China emission inventories. All authors (Z.M., J.M., D.G., Y.S., M.S., Y.-M.W., Z.L. and K.H.) participated in the writing of the manuscript. D.G. coordinated and supervised the project

## Additional information

**Competing interests:** The authors declare no competing financial interests.

