## [Peer Review File · Nature Communications]

Reviewer #1 (Remarks to the Author):

This manuscript adopted a multi-regional input-output model to analyze the interregional carbon emission flows in China after the financial crisis (the 'new normal' era). The results are clearly presented and discussed. The main message that emission flow patterns have changed greatly as the (interregional and international) trade patterns changed is straightforward but solid. Some prospective discussion was also provided based on these results.

My main concern on its suitability in Nature Communications is the novelty and significance of this manuscript. Method wise, the past decade has seen a great amount of work on the development and use of multi-regional input-output analysis for consumption-based emission accounting (including the authors' own work). A potential innovation in this manuscript is its consideration of interregional trade within China; however, previous papers have included this development and its application in understanding China's inter-provincial carbon leakage, for example, Feng, K., Davis, S.J., Sun, L., Li, X., Guan, D., Liu, W., Liu, Z., Hubacek, K., 2013. Outsourcing CO₂ within China. Proc. Natl. Acad. Sci. 100, 11654–11659 (By the way, this paper of the authors' own was NOT cited in the manuscript). Of course, the manuscript has used data of 2007 and 2012 (while Feng et al. 2013 used data of 2007 only), and thus can reveal changes after the financial crisis. But the changes are relatively straightforward (the way of presenting data in Figure 1 and 2 is the same as that in Feng et al. 2013) and has been previously touched in other work, for example, Peters, G.P., Marland, G., Le Quéré, C., Boden, T., Canadell, J.G., Raupach, M.R., 2011. Rapid growth in CO₂ emissions after the 2008-2009 global financial crisis. Nat. Clim. Chang. 2, 2–4 (though conclusion is different since Peters et al 2011 included only the years 2008 and 2009).

Some other minor comments:

- I think territory-based emission is more suitable than production-based emission.
- Line 343-345: Do you think the increase trend from 2010 to 2012 will continue and thus bring the emissions back to the pre-crisis levels in the future?
- Ending paragraph: While I agree with the direction that carbon leakage in south-south trade deserves more attention, I personally find these sentences as concluding remarks a bit too strong (the majority of carbon leakage would still occur between developed and developing countries in the near future).

Reviewer #2 (Remarks to the Author):

The authors constructed the multi-country/multi-region I-O tables by linking Chinese multi-region I-O tables and global multi-country I-O tables (GTAP dataset) in 2012 to study the embodied emission flows in China. They provide the latest findings regarding to the embodiment in interregional and international trade of 30 regions in China after global financial crisis. The authors further compared the 2012 results with results for year 2007 and 2010. This is the positive contribution of this paper. There are also some parts need further improvements, including missing important references in the literature, some unclear parts in the data treatment, and results comparisons. The major and specific comments are given below:

The unique contribution of this paper comes from the linking the Chinese regional I-O tables and global I-O tables using the latest dataset for year 2012. Similar linkages have been done by previous studies on earlier years, such as work by Feng et al. (2013; PANS 110, pp.11654-11659), Su and Ang (2014; Applied Energy 114, pp.377-384) and Weitzel and Ma (2014; Energy Economics 45, pp.45-52). The authors are suggested to include them into the discussions.

Besides, direct comparison of the embodied emissions results between 2012 and 2007/2010 is not sufficient to understand what caused such differences. The driving forces can include energy efficiency improvement, energy mix changes, emission coefficient change, production structure change, demand structure change and total demand change, which can be obtained through

applying the structural decomposition analysis (SDA). See the review of SDA studies on energy and emissions in Su and Ang (2012; Energy Economics 34, pp.177-188; for studies published before 2010) and Wang et al. (2017; Energy Policy 107, 585-599; for studies published in 2010-2015). The authors are suggested to conduct the SDA to provide the insights into the embodied emission changes in 2007-2010 and 2010-2012.

Some specific comments:

Lines 30-31, "shift from an investment-driven economy to a consumption-driven economy": Recent work by Su and Ang (2017; Energy Economics 65, 137-147)'s Table 2 shows that the investment in China accounts for 43.5% ($=2664.0/6080.7$) of its total emissions in 2007 and 52.7% ($=4312/8185.4$) of its total emissions in 2012. I don't think this statement is correct.

Lines 42-43: The authors are suggested to include other important references, such as Su and Ang (2014; Applied Energy 114, pp.377-384) and Weitzel and Ma (2014; Energy Economics 45, pp.45-52).

Lines 59-60: The authors should read the work by Su and Thomson (2016; Energy Economics 59, 414-422) who give the time-series estimates (2006-2012) of China's emissions embodied in both processing and normal exports and identify the driving forces to the embodied emission changes through SDA. Their studying period covers the changes before and after global financial crisis in 2008-2009.

Lines 61-63: From Table 2 in Su and Ang (2017; Energy Economics 65, 137-147), the share of value added embodied in China's exports dropped from 26.5% ($=6,999.7/26,481.3$) in 2007 to 20.9% ($=9883.8/47384.9$) in 2012. Their conclusion is different from this statement.

Section "Discussions": The driving factors of the embodied emission changes between 2007 and 2012 are not clear. The authors are suggested to apply the SDA to investigate these driving factors.

Lines 340-341, "emissions embodied in China's foreign exports have peaked": The authors are suggested to compare the time-series estimates (2006-2012) of China's emissions embodied in exports reported in Su and Thomson (2016; Energy Economics 59, 414-422), see their Figure 2 and Table 5&6. It cannot conclude that the embodied have peaked.

Table S1: The production-based emissions by 30 regions in China reported in this table are different from the supporting tables attached in the excel files. For example, Beijing's production-based emissions are 88 MT/86 MT/80 MT in 2007/2010/2012 in Table S1, but the excel files show that the estimates are 102.97MT/102.97MT/97.19MT in the excel files. Please check which datasets are correct.

Section "Methods": I am not sure whether the authors compiled only the 2012 Chinese Multi-region I-O tables, or all three multi-region tables (2007, 2010, 2012). The 2007 and 2010 Chinese Multi-region I-O tables (30 region and 30 sectors) have already been compiled and published by Prof. Liu Weidong's group in the Institute of Geographic Sciences and Natural Resource Research, Chinese Academic of Sciences. Did the authors directly use their published tables or compiled by themselves?

Lines 473-474, "we adopt the 2011 GTAP model to the years 2010 and 2012 xxx": Have you updated the economic/emission performances of the world economics to the year 2010 and 2012 in the study? If yes, what are the data sources used in the update?

Lines 480-481: Please provide the matching table between the 57-sector classification in GTAP and 30-sector classification in Chinese multi-region IO tables.

Lines 482-484: Why don't the authors use the same number of world countries in the different years' analysis? Besides, what are the assumptions and data sources used for the Rest of the World (ROW)? For example, 11 regions captured in 2010 and 2012 are included in the ROW.

Lines 525-533: As mentioned earlier, the sources of the Chinese multi-region I-O tables for 2007 and 2010 are not clear.

Reviewer #3 (Remarks to the Author):

Previous studies have mapped and quantified the material flows and GHGE generation consumption and production relationship between China and the rest of the world. This study goes further: it untangles the differences in production and consumption based accounting methods in the regions of China over a time series (2007-2012, during the GFC), providing valuable insights. Specifically on the domestic and international outsourcing of emissions. It shows that the outsourcing of emissions is not just a Developed->Developing world problem, but also a problem between sub regions of developing countries, and also between developing countries. This work is rigorous, original and very publishable. It should attract scholarly (and media attention), it will influence thinking in the field (and hopefully broader policy debate around sub-regional development with regards to carbon intensive industries). Given the methods and appendices provided, this study findings should be reproducible.

Some brief comments:

Line 124-151

The functional unit of choice for this paper is g of c02e per yuan, per year or per capita. However these expressions of carbon relationship may be masking material and physical flows, as there is little discussion of the types of goods exported (east vs west), ie the choice of functional units may hide a decoupling the relationship of high cost goods vs high carbon intensive goods. With a 30 sector IOT (with uncertain data) there is little chance for further exploration of the exact goods traded but this would be an interesting further investigation. (although I also understand there is little room to go deeply into this relationship in a nature comms article)

437-451 impact exponents and inter regional cooperation. I would like to see a worked through example of this method and results as a separate article using the data set in this paper, I understand the space limitations of a NCC article and feel this additional article could be sent to ESR or similar.

478-492 Could the authors provide further explanation as to why they chose to aggregate 30 by 30 rather than dis-aggregating to 57 by 57? As this would have provided more detail of some of the trade linkages.

Response to Reviewer #1:

This manuscript adopted a multi-regional input-output model to analyze the interregional carbon emission flows in China after the financial crisis (the 'new normal' era). The results are clearly presented and discussed. The main message that emission flow patterns have changed greatly as the (interregional and international) trade patterns changed is straightforward but solid. Some prospective discussion was also provided based on these results.

Response:

Many thanks for your support and valuable comments. We have carefully revised the manuscript following your comments.

My main concern on its suitability in Nature Communications is the novelty and significance of this manuscript. Method wise, the past decade has seen a great amount of work on the development and use of multi-regional input-output analysis for consumption-based emission accounting (including the authors' own work). A potential innovation in this manuscript is its consideration of interregional trade within China; however, previous papers have included this development and its application in understanding China's inter-provincial carbon leakage, for example, Feng, K., Davis, S.J., Sun, L., Li, X., Guan, D., Liu, W., Liu, Z., Hubacek, K., 2013. Outsourcing CO₂ within China. *Proc. Natl. Acad. Sci.* 100, 11654 - 11659 (By the way, this paper of the authors' own was NOT cited in the manuscript). Of course, the manuscript has used data of 2007 and 2012 (while Feng et al. 2013 used data of 2007 only), and thus can reveal changes after the financial crisis. But the changes are relatively straightforward (the way of presenting data in Figure 1 and 2 is the same as that in Feng et al. 2013) and has been previously touched in other work, for example, Peters, G.P., Marland, G., Le Quéré, C., Boden, T., Canadell, J.G., Raupach, M.R., 2011. Rapid growth in CO₂ emissions after the 2008-2009 global financial crisis. *Nat. Clim. Chang.* 2, 2 - 4 (though conclusion is different since Peters et al 2011 included only the years 2008 and 2009).

Response:

Many thanks for your comments. During the first round submission, we made clear to the editorial team that our study is based on a widely used approach that provides robust results and that the contributions are the new dataset that was developed and the relevant policy analysis and new insights that can be achieved with the new dataset. We also made this point clearer in revised manuscript.

We utilized the latest socioeconomic datasets to compile 2012 China's multiregional input-output (MRIO) table. This is a new data contribution to the academic field. All MRIO tables developed and used in this study are provided as Supplementary Data for this submission, which will be made publicly available via data repository after the publishing process.

We estimated the carbon implications of recent changes in China's economic development patterns and role in global trade in the post-financial-crisis era (i.e. 2007-2012). Compared with previous analysis, we discovered an interesting reversal of emission flows between Chinese regions and some important changes globally. In addition, as suggested by Reviewer 2, we added a structural decomposition analysis (SDA), based on the latest available data, to analyze driving forces behind these emission changes. Seven factors were included in the analysis: emission coefficient, energy mix, energy efficiency, production structure, consumption structure, consumption per capita, and population. The results show that the shift of Southwest China from being a net emission exporter to being a net emission importer was mainly due to the rapid growth in consumption in these regions. The decline of China's export emissions from 2007 to 2012 was mainly due to production structure changes and efficiency gains. The SDA results on emissions embodied in China's domestic and foreign trade are shown in Figure 2 and Figure 4c, respectively.

The two papers you mentioned have now been added in the revised manuscript (reference 6 and 17).

Some other minor comments:

- I think territory-based emission is more suitable than production-based emission.

Response:

Many thanks. All "production-based emission" in the manuscript has been replaced by "territory-based emission".

- Line 343-345: Do you think the increase trend from 2010 to 2012 will continue and thus bring the emissions back to the pre-crisis levels in the future?

Response:

We think that China's export emissions will not return to the 2007 level for two arguments. First, China's export volume has been decreasing since 2012. Second, China's production structure is becoming less carbon emission intensive.

- Ending paragraph: While I agree with the direction that carbon leakage in south-south trade deserves more attention, I personally find these sentences as concluding remarks a bit too strong (the majority of carbon leakage would still occur between developed and developing countries in the near future).

Response:

We followed your suggestions and deleted related statements in the ending paragraph. We agree with you that the majority of carbon leakage would still occur between developed and developing countries in the near future.

Response to Reviewer #2:

The authors constructed the multi-country/multi-region I-O tables by linking Chinese multi-region I-O tables and global multi-country I-O tables (GTAP dataset) in 2012 to study the embodied emission flows in China. They provide the latest findings regarding to the embodiment in interregional and international trade of 30 regions in China after global financial crisis. The authors further compared the 2012 results with results for year 2007 and 2010. This is the positive contribution of this paper. There are also some parts need further improvements, including missing important references in the literature, some unclear parts in the data treatment, and results comparisons. The major and specific comments are given below:

Response:

Many thanks for your support and valuable comments. We have carefully revised the manuscript following your comments.

The unique contribution of this paper comes from the linking the Chinese regional I-O tables and global I-O tables using the latest dataset for year 2012. Similar linkages have been done by previous studies on earlier years, such as work by Feng et al. (2013; PANS 110, pp.11654-11659), Su and Ang (2014; Applied Energy 114, pp.377-384) and Weitzel and Ma (2014; Energy Economics 45, pp.45-52). The authors are suggested to include them into the discussions.

Response:

We have followed your suggestions and added and summarized the three studies (line 487-490).

Besides, direct comparison of the embodied emissions results between 2012 and 2007/2010 is not sufficient to understand what caused such differences. The driving forces can include energy efficiency improvement, energy mix changes, emission coefficient change, production structure change, demand structure change and total demand change, which can be obtained through applying the structural decomposition analysis (SDA). See the review of SDA studies on energy and emissions in Su and Ang (2012; Energy Economics 34, pp.177-188; for studies published before 2010) and Wang et al. (2017; Energy Policy 107, 585-599; for studies published in 2010-2015). The authors are suggested to conduct the SDA to provide the insights into the embodied emission changes in 2007-2010 and 2010-2012.

Response:

We have followed your suggestion and applied SDA to estimate the driving forces of embodied emission changes in 2007-2010 and 2010-2012. Seven factors were considered, including emission coefficient, energy mix, energy efficiency, production structure, consumption structure, consumption per capita, and population. The SDA results are shown in Figure 2 and Figure 4c. We added an introduction on SDA approach in the methods section. The two review papers were added in the revised manuscript (line 547-549).

Some specific comments:

Lines 30-31, “shift from an investment-driven economy to a consumption-driven economy” : Recent work by Su and Ang (2017; Energy Economics 65, 137-147)’ s Table 2 shows that the investment in China accounts for 43.5% (=2664.0/6080.7) of its total emissions in 2007 and 52.7% (=4312/8185.4) of its total emissions in 2012. I don’ t think this statement is correct.

Response:

We have deleted the statement related to the “shift from an investment-driven economy to a consumption-driven economy”. The useful references you provided have been added (reference 4).

Lines 42-43: The authors are suggested to include other important references, such as Su and Ang (2014; Applied Energy 114, pp.377-384) and Weitzel and Ma (2014; Energy Economics 45, pp.45-52).

Response:

The two important references have been included (line 40-42, reference 7 and 8).

Lines 59-60: The authors should read the work by Su and Thomson (2016; Energy Economics 59, 414-422) who give the time-series estimates (2006-2012) of China’ s emissions embodied in both processing and normal exports and identify the driving forces to the embodied emission changes through SDA. Their studying period covers the changes before and after global financial crisis in 2008-2009.

Response:

We have read the work of Su and Thomson (2016) and added related discussions based on this work (line 58-61, reference 10).

Lines 61-63: From Table 2 in Su and Ang (2017; Energy Economics 65, 137-147), the share of value added embodied in China’s exports dropped from 26.5% (=6,999.7/26,481.3) in 2007 to 20.9% (=9883.8/47384.9) in 2012. Their conclusion is different from this statement.

Response:

The definitions of share of value added embodied in exports are different. Su and Ang (2017) discussed the value added embodied in different final demand categories, including rural consumption, urban consumption, government consumption, gross fixed capital formation, inventory change, and exports. In our study, we discussed the domestic and foreign value added in exports. The definitions in Su and Ang (2017) and our study are shown in the following equations.

$$\text{Share in Su and Ang (2017)} = \frac{\text{Value added embodied in exports}}{\text{Value added embodied in final demand}}$$

$$\text{Share in our paper} = \frac{\text{Domestic value added in exports}}{\text{Value added in exports}}$$

Section “Discussions” : The driving factors of the embodied emission changes between 2007 and 2012 are not clear. The authors are suggested to apply the SDA to investigate these driving factors.

Response:

We have applied SDA to investigate the driving forces behind the embodied emission changes.

Lines 340-341, “emissions embodied in China’s foreign exports have peaked” : The authors are suggested to compare the time-series estimates (2006-2012) of China’s emissions embodied in exports reported in Su and Thomson (2016; Energy Economics 59, 414-422), see their Figure 2 and Table 5&6. It cannot conclude that the embodied have peaked.

Response:

We have deleted the statement on the peak of emissions embodied in exports. The change trends on emissions embodied in China’s exports in our study are consistent with those in Su and Thomson (2016). We compared our results with those of Su and Thomson (2016) (line 250-252).

Table S1: The production-based emissions by 30 regions in China reported in this table are different from the supporting tables attached in the excel files. For example, Beijing’ s production-based emissions are 88 MT/86 MT/80 MT in 2007/2010/2012 in Table S1, but the excel files show that the estimates are 102.97MT/102.97MT/97.19MT in the excel files. Please check which datasets are correct.

Response:

We have made emission data consistent between Table S1 and excel files. In our original submission, the data in excel files included emissions from economic sectors and residential energy consumption, while Table S1 only included emissions from economic sectors. That’s why the overall value was different. In the new submission, we deleted the emissions from residential energy consumption in the excel files.

Section “Methods” : I am not sure whether the authors compiled only the 2012 Chinese Multi-region I-O tables, or all three multi-region tables (2007, 2010, 2012). The 2007 and 2010 Chinese Multi-region I-O tables (30 region and 30 sectors) have already been compiled and published by Prof. Liu Weidong’ s group in the Institute of Geographic Sciences and Natural Resource Research, Chinese Academic of Sciences. Did the authors directly use their published tables or compiled by themselves?

Response:

We only compiled the 2012 Chinese MRIO tables. The 2007 and 2010 Chinese MRIO tables are compiled by in the Institute of Geographic Sciences and Natural Resource Research, Chinese Academy of Sciences. We added a statement in the methods section (line 571-573). Our method for compiling 2012 MRIO table is based on the modified gravity model, which is the same as that for 2007 and 2010 MRIO tables.

Therefore, our 2012 MRIO table is consistent and comparable with the 2007 and 2010 tables.

Lines 473-474, “we adopt the 2011 GTAP model to the years 2010 and 2012 xxx” : Have you updated the economic/emission performances of the world economics to the year 2010 and 2012 in the study? If yes, what are the data sources used in the update?

Response:

We updated the economic and emission performance of the world economics to the year 2010 and 2012. The GDP data was obtained from the National Accounts Main Aggregates Database, and the CO₂ emissions were obtained from the Carbon Dioxide Information Analysis Center (CDIAC). We added the related statements and data sources in the methods section (line 577-580).

Lines 480-481: Please provide the matching table between the 57-sector classification in GTAP and 30-sector classification in Chinese multi-region IO tables.

Response:

We added Table S5 to show the concordance of sectors for GTAP and China MRIO.

Lines 482-484: Why don't the authors use the same number of world countries in the different years' analysis? Besides, what are the assumptions and data sources used for the Rest of the World (ROW)? For example, 11 regions captured in 2010 and 2012 are included in the ROW.

Response:

The differences in number of world regions in different years are determined by the GTAP database which we use in this study. The world is divided into 129 regions in the 2007 GTAP database and 140 regions in the 2011 GTAP database. In our discussions, we aggregate the world into 11 regions (see Table S2 for the detailed definition of world regions).

The assumptions and data for the Rest of the World (ROW) are obtained from the GTAP database. Introduction of the regional disaggregation can be found on the GTAP website (<https://www.gtap.agecon.purdue.edu/databases/regions.asp?Version=9.211>). The ROW includes: Antarctica, Bouvet Island, British Indian Ocean Territory, and French Southern Territories.

Lines 525-533: As mentioned earlier, the sources of the Chinese multi-region I-O tables for 2007 and 2010 are not clear.

Response:

The 2007 and 2010 Chinese MRIO tables are compiled by the Institute of Geographic Sciences and Natural Resource Research, Chinese Academic of Sciences. We added statements and references in the methods section (line 571-573).

Response to Reviewer #3:

Previous studies have mapped and quantified the material flows and GHGE generation consumption and production relationship between China and the rest of the world. This study goes further: it untangles the differences in production and consumption based accounting methods in the regions of China over a time series (2007-2012, during the GFC), providing valuable insights. Specifically on the domestic and international outsourcing of emissions. It shows that the outsourcing of emissions is not just a Developed->Developing world problem, but also a problem between sub regions of developing countries, and also between developing countries. This work is rigorous, original and very publishable. It should attract scholarly (and media attention), it will influence thinking in the field (and hopefully broader policy debate around sub-regional development with regards to carbon intensive industries). Given the methods and appendices provided, this study findings should be reproducible.

Response:

Many thanks for your support and valuable comments. We have carefully revised the manuscript following your comments.

Some brief comments:

Line 124-151

The functional unit of choice for this paper is g of c02e per yuan, per year or per capita. However these expressions of carbon relationship may be masking material and physical flows, as there is little discussion of the types of goods exported (east vs west), ie the choice of functional units may hide a decoupling the relationship of high cost goods vs high carbon intensive goods. With a 30 sector IOT (with uncertain data) there is little chance for further exploration of the exact goods traded but this would be an interesting further investigation. (although I also understand there is little room to go deeply into this relationship in a nature comms article)

Response:

Many thanks for your comments. We agree with you and are aware that the exact goods traded among Chinese regions (east vs west) are an interesting topic. For this research, we could guarantee the method and data robustness of a 30x30 sector MRIO. This is also comparable with earlier MRIO format and compilation processes. We will consider further investigation on different types of goods traded among Chinese regions in future research, but currently this is beyond the scope of the presented study. Most trade studies based on MRIO share this limitation due to available MRIOs and data but at the same have the advantage of comparability of results.

437-451 impact exponents and inter regional cooperation. I would like to see a worked through example of this method and results as a separate article using the data set in this paper, I understand the space limitations of a NCC article and feel this additional article could be sent to ESR or similar.

Response:

We have added a worked though example of this method in Supplementary Information. We take agriculture sector among 10 Chinese regions as an example to demonstrate the modification process.

Many thanks for your suggestions on the additional article to ESR. We will make all compiled data publicly available via data repository. After communicated with the Nature Communication editor, we may prepare such method and data paper and submit to *Scientific Data*. Nevertheless, we place a working example and relevant datasets in the Supplementary Information so that readers can get access to available data and method related to this research.

478-492 Could the authors provide further explanation as to why they chose to aggregate 30 by 30 rather than dis-aggregating to 57 by 57? As this would have provided more detail of some of the trade linkages.

Response:

We are aware that 57-sector MRIO would provide more detail of trade linkages. For this research, we could guarantee the method and data robustness of a 30x30 sector MRIO. This is also comparable with earlier MRIO format and compilation processes. The main purpose of this paper is to compare China's emission flows from 2007 to 2012. So we aggregate 57 sectors into 30 sectors to make the 2012 MRIO model comparable with those for 2010 and 2007. We will consider a further disaggregation for IO sectors, but currently this is beyond the scope of the presented study.

Reviewer #2 (Remarks to the Author):

The paper has been significantly improved after the revision. Quite valuable results have been illustrated regarding to the embodied emission flows changes at both international and interregional levels and associated driving forces behind such changes in 2007-2012. This study provides the fresh insights into the embodiment situation changes before and after the global financial crisis. Such timely results can attract readers from different fields. I don't have further comments to address and highly recommend it for publication in this journal.

Reviewer #3 (Remarks to the Author):

I find the responses to my (and the other authors) queries and notes satisfactory

Point-by-point response to the referees' comments

Reviewer #2 (Remarks to the Author):

The paper has been significantly improved after the revision. Quite valuable results have been illustrated regarding to the embodied emission flows changes at both international and interregional levels and associated driving forces behind such changes in 2007-2012. This study provides the fresh insights into the embodiment situation changes before and after the global financial crisis. Such timely results can attract readers from different fields. I don't have further comments to address and highly recommend it for publication in this journal.

Response: Thank you very much for your comments and support.

Reviewer #3 (Remarks to the Author):

I find the responses to my (and the other authors) queries and notes satisfactory

Response: Thank you very much for your comments and support.